# Depth of the Steroid Core Location Determines the Mode of Na,K-ATPase Inhibition by Cardiotonic Steroids

**DOI:** 10.3390/ijms222413268

**Published:** 2021-12-09

**Authors:** Artem M. Tverskoi, Yuri M. Poluektov, Elizaveta A. Klimanova, Vladimir A. Mitkevich, Alexander A. Makarov, Sergei N. Orlov, Irina Yu. Petrushanko, Olga D. Lopina

**Affiliations:** 1Engelhardt Institute of Molecular Biology, Russian Academy of Sciences, 32 Vavilova Street, 119991 Moscow, Russia; yuripoul@gmail.com (Y.M.P.); mitkevich@gmail.com (V.A.M.); aamakarov@eimb.ru (A.A.M.); irina-pva@mail.ru (I.Y.P.); 2Faculty of Biology, Lomonosov Moscow State University, 1/12 Leniskie Gory Street, 119234 Moscow, Russia; klimanova.ea@yandex.ru

**Keywords:** cardiotonic steroids, Na,K-ATPase, inhibition mode, structural changes, cardiotonic steroids-binding site

## Abstract

Cardiotonic steroids (CTSs) are specific inhibitors of Na,K-ATPase (NKA). They induce diverse physiological effects and were investigated as potential drugs in heart diseases, hypertension, neuroinflammation, antiviral and cancer therapy. Here, we compared the inhibition mode and binding of CTSs, such as ouabain, digoxin and marinobufagenin to NKA from pig and rat kidneys, containing CTSs-**s**ensitive (α1**S**) and -**r**esistant (α1**R**) α1-subunit, respectively. Marinobufagenin in contrast to ouabain and digoxin interacted with α1S-NKA reversibly, and its binding constant was reduced due to the decrease in the deepening in the CTSs-binding site and a lower number of contacts between the site and the inhibitor. The formation of a hydrogen bond between Arg111 and Asp122 in α1R-NKA induced the reduction in CTSs’ steroid core deepening that led to the reversible inhibition of α1R-NKA by ouabain and digoxin and the absence of marinobufagenin’s effect on α1R-NKA activity. Our results elucidate that the difference in signaling, and cytotoxic effects of CTSs may be due to the distinction in the deepening of CTSs into the binding side that, in turn, is a result of a bent-in inhibitor steroid core (marinobufagenin in α1S-NKA) or the change of the width of CTSs-binding cavity (all CTSs in α1R-NKA).

## 1. Introduction

Cardiotonic steroids (CTSs) or cardiac glycosides are a group of steroid compounds originally found in plants that were used for the treatment of heart failure. Later, they were also found in the skin of toads, and some CTSs were isolated from human and animal biological liquids (endogenous CTSs). At least three CTSs (ouabain, marinobufagenin and telocinobufagin) were found in human plasma and/or urine [1,2,3,4]; besides that, several digoxin-immunoreactive compounds were also revealed [5,6]. It was demonstrated that the increase in the endogenous CTSs concentration in human and animal blood may be involved in the development of preeclampsia, chronic kidney disease, hypertension and other cardiovascular diseases [7,8,9,10,11,12]. Changes in the marinobufagenin level can affect both vessel cells and blood cells, notably altering the Na,K-ATPase activity of erythrocytes [13]. Interest in different CTSs is also induced by their application in the treatment of cardiac diseases and potential use as agents with recently discovered anticancer [14] and antiviral activity [15,16,17,18,19,20,21,22]. Recently, it was found that CTSs can reduce neuroinflammation in a mouse model of Alzheimer’s disease [23]. All CTSs share a common structure: they have a steroid core with four *cis*-*trans*-*cis*-fused rings, a lactone ring at the 17-th position and a hydroxyl group at C-14. The five- and six-membered lactone rings are characteristic for cardenolides and bufadienolides, respectively (Appendix A).

CTSs are highly specific inhibitors of Na,K-ATPase (NKA) known as Na-pump. NKA is located in the plasma membrane of animal cells and transports three Na^+^ ions from the cell and two K^+^ ions into the cell against an electrochemical gradient utilizing the energy of ATP hydrolysis [24]. NKA is composed of a large catalytic α-subunit and a regulatory β-subunit. Four isoforms of α-subunits (α1–α4) and three of β-subunits (β1–β3) are expressed in mammals. The α1-isoform is expressed in all types of animal cells. Isoforms of both subunits can form dimer in various combinations. Depending on the combination of α- and β-isoforms, different NKA isozymes have various affinities to ATP, Na^+^, K^+^ and ouabain, which is the best-studied cardiotonic steroid. However, the defining role in the change of the affinity to ouabain is played by the α-isoform of the enzyme (for review, see [25,26,27,28,29,30]).

It is known that the affinity of the α1-NKA isoform for ouabain and other CTSs in rodents is about 1000-fold lower (**r**esistant α1**R**-NKA) than that in other mammals (**s**ensitive α1**S**-NKA). CTS-resistance of α1R-NKA in rodents is mainly due to the substitution of uncharged amino acid residues Gln111 and Asn122 with charged Arg and Asp [31]. These amino acids are located near the extracellular surface of the membrane at the end of the first and at the start of the second transmembrane alpha-helices (M1 and M2) in proximity.

Ouabain in concentrations that completely inhibit the transport function of NKA (100 nM–10 µM) triggered the death of various cells with α1S-NKA [32,33,34,35,36,37,38,39,40,41,42,43,44,45,46,47,48,49]. In contrast, treatment of rodent cells containing α1R-NKA with ouabain (1–3 mM), which inhibits the transport function of the enzyme, did not affect the survival [45,50,51,52,53]. Furthermore, the transfection of α1R-NKA in renal epithelial cells from the canine kidney (MDCK) and human umbilical vein endothelial cells (HUVEC) protected cells from the ouabain-induced death (1–3 mM of ouabain) [45,54], and this effect did not cause NKA inhibition and attenuation of the [Na^+^]_i_/[K^+^]_i_ ratio. In addition, unlike mice cells with wild-type α1^R/R^-NKA, 3 µM of ouabain-triggered death of mice cells that expressed human α1^S/S^-NKA [45]. Earlier, it was shown that NKA is not only an ion pump but also a receptor, activating several signaling pathways via conformational transitions of the NKA α-subunit and changing cell behavior [55,56,57,58,59,60]. Later, it was shown that ouabain in concentrations completely inhibiting NKA led to the phosphorylation of p38 mitogen-activated protein kinase (MAPK) and the death of HUVEC (contain α1S-NKA), whereas, in rat aorta endothelial cells (RAEC; contain α1R-NKA), it stimulated the phosphorylation of ERK1/2 MAPK and did not affect cell viability [45]. Hence, the presence of α1R-NKA protects cells from the cytotoxic action of ouabain.

Diverse CTSs in mammals imply their various functional roles [61,62]. Indeed, there is a significant difference between the therapeutic effects of ouabain and digoxin [63]. Simultaneously, in a number of pathologies, the change in the endogenous marinobufagenin concentration is more pronounced in comparison with ouabain. It is still an open question on the difference in the action of diverse CTSs, for instance, marinobufagenin and ouabain/digoxin. It was shown earlier that marinobufagenin induced renal epithelial cells death at higher concentrations than ouabain (about 1 and 0.1 μM, respectively) despite inhibition of ion transport by NKA in the cells by 50% at these concentrations [34]. Moreover, we observed a 17-fold higher affinity for the ouabain binding to E2P-conformation of NKA in comparison to marinobufagenin (K_d_ values were equal to 0.1 and 1.7 μM, respectively) that were bound to the same site of NKA from duck salt glands [64] (α1β1-isoform [65]). At the same time, marinobufagenin, unlike ouabain, interacts with the E1-conformation of NKA [64]. Ouabain and marinobufagenin binding to NKA leads to various structural changes in enzymes [64]. Therefore, different CTSs induce diverse physiological effects, for example, on cell survival, heart muscle metabolism and blood pressure regulation, but the underlying mechanisms remain unknown.

In the present study, we compared binding parameters and inhibition mode of CTSs to NKA from pig kidneys (α1S-NKA) and rat kidneys (α1R-NKA). Three CTSs were used in this study: ouabain, digoxin (cardenolides) and marinobufagenin (bufadienolide). Using data from X-ray analysis of NKA from pig kidney [66,67], we created models of CTSs-binding sites for α1S- and α1R-NKA and determined the differences in the location of ouabain, digoxin and marinobufagenin in them. Our results show that the distinction in deepening of CTSs determines the differences between the binding parameters and the mode of NKA inhibition by diverse CTSs. The data are useful for the understanding of our previous findings of the discrepancy between NKA conformational changes induced by various CTSs [68] as well as between cytotoxic CTSs action on human and rodent cells [45].

## 2. Results

### 2.1. Interactions of Cardiotonic Steroids with α1S- and α1R-Na,K-ATPase

It was established earlier that CTSs bind to NKA with a higher affinity in E2P-conformation [69]. We used isothermal titration calorimetry (ITC) for direct determination of the thermodynamic parameters for the binding of ouabain, digoxin and marinobufagenin to α1S- and α1R-NKA in E2P-conformation (Table 1). A typical set of ITC data for CTSs binding to α1-NKAs in E2P-conformation at 37 °C is shown in Figure 1.

We observed a decrease in the affinity for the binding of α1S-NKA to CTSs in the range of ouabain, digoxin and marinobufagenin (K_d_ values are equal to 53, 208 and 2320 nM, respectively). Complex formation between CTSs and α1S-NKA is an enthalpy favorable process (Table 1); however, in the case of ouabain, the enthalpy contribution (ΔH) to the energetics of complex formation is very high (−21.6 ± 0.7 kcal/mol), and in the case of digoxin and marinobufagenin, it significantly decreases (−7.9 ± 0.6 and −5.2 ± 0.3 kcal/mol, respectively), and the contribution of the entropy component (−TΔS) increases (−2.0 and −2.8 kcal/mol, respectively).

In contrast to α1S-NKA, there was no measurable binding of α1R-NKA to digoxin and marinobufagenin. Ouabain affinity for α1R-NKA is 60 times lower than for α1S-NKA (Table 1). Energetics of complex formation between ouabain and α1R-NKA also significantly differs from that for α1S-NKA. The contribution of the entropy and enthalpy components to the change in the Gibbs energy upon binding becomes practically the same (ΔH = −4.3 ± 0.7 kcal/mol; −TΔS = −3.4 kcal/mol).

### 2.2. Inhibition of α1S- and α1R-Na,K-ATPase by Cardiotonic Steroids

The dependencies of hydrolytic activity of α1S- and α1R-NKA upon the concentration of ouabain, digoxin and marinobufagenin are shown in Figure 2A,B. Curves describing α1S-NKA inhibition by these CTSs are slightly different, but all of them fit to the hyperbola with close IC_50_ values (0.8, 1.2 and 2.0 μM for marinobufagenin, digoxin and ouabain, respectively). Inhibition of rat enzyme by ouabain and digoxin was also described by the hyperbola with the values of IC_50_ equal 140 and 250 μM. Marinobufagenin in a concentration up to 500 μM does not inhibit α1R-NKA.

The dependencies of α1S-NKA (E2P-conformation) activity upon ouabain concentrations in the preincubation medium at different enzyme concentrations (40, 80 and 400 nM) are shown in Figure 2C. All obtained curves fit to sigmoid with Hill coefficient close to 2 (1.89 ± 0.14, 1.96 ± 0.19 and 1.91 ± 0.09 for 40, 80 and 400 nM of the enzyme, respectively). The dependencies of pig kidney α1S-NKA activity on digoxin concentrations in the preincubation medium (E2P-conformation) at different enzyme concentrations are presented in Figure 2D. They look similar to those observed for ouabain inhibition. Curves obtained also fit to sigmoid with Hill coefficient close to 2 (1.90 ± 0.21, 1.70 ± 0.33 and 1.99 ± 0.27 for 40, 80 and 400 nM of the enzyme, respectively). It demonstrates that ouabain and digoxin are pseudo-irreversible inhibitors of α1S-NKA, and they bind to two sites of the enzyme in E2P-conformation with positive cooperative interactions between them.

We did not find any inhibition of α1S-NKA after the preincubation with marinobufagenin; hence, this CTS may be considered a reversible inhibitor of α1S-NKA. Preincubation of α1R-NKA from rat kidney with ouabain does not result in enzyme inhibition in comparison with the corresponding control. It shows that ouabain interacts with α1R-NKA reversibly. Similar results were obtained with digoxin and marinobufagenin.

### 2.3. Docking of Cardiotonic Steroids with α1R- and α1S-Na,K-ATPase

As mentioned above, the CTS-resistance of α1R-NKA in rodents is mainly due to the substitution of two uncharged amino acid residues (Gln111 and Asn122) with charged amino acids (Arg and Asp) [31]. They are located close to the extracellular surface near the entry into the channel of the CTSs-binding site. Taking this into account, we created a model of CTSs-binding site for α1R-NKA based on the previously published 3.4 Å structure of the porcine α-subunit α1S-NKA in E2P-conformation in complex with ouabain (PDB code 4HYT) and digoxin (PDB code 4RET).

After substitution of indicated amino acid residues in 4RET and 4HYT structures, the penetrability of the channel formed by M1-M5 was assessed using its ability to interact with CTSs correctly. It was found that Arg111 and Asp122 form a hydrogen bond and thus create steric hindrance for entry into the channel of the CTSs-binding site (Figure 3A).

To validate the model, we removed CTSs from the native structures of porcine NKA, created models and made docking with constructed CTSs. In both cases, docking with the minimized model of α1S-NKA results in the positioning of ouabain into the channel formed by M1–M5 helixes like ouabain from structure 4HYT (root mean square deviation (RMSD)—0.57 Å for the lactone ring and steroid core), and a molecule of digoxin was in the channel similar to the ligand from structure 4RET (RMSD—0.73 Å for the lactone ring and steroid core). The difference in the deepening of the CTSs’ steroid core between native structures and models did not exceed 0.5 Å (Figure 3B,C).

To establish the role of Gln111Arg and Asn122Asp substitutions in the protein interactions with CTSs, we performed docking of ouabain (OBN), digoxin (DGX), and marinobufagenin (MBG) with models of rat α1R- and pig α1S-NKA. We compared the positioning of different CTSs in the CTSs-binding sites of α1S- and α1R-NKA (Figure 3E,F and Figure 4D,E).

We found a very small difference between the location of digoxin and ouabain in a complex of CTSs-sensitive α1S-NKA (4HYT; the difference between the deepening of C23 of digoxin and ouabain did not exceed 0.7 Å). However, a more superficial location was characteristic of marinobufagenin. The distance between C23 of paired digoxin and marinobufagenin, as well as of paired ouabain and marinobufagenin, consists of approximately 5 Å (Figure 4 and Appendix A). Similar results were obtained for the structure of 4RET.

According to our data, the ligands interacting with the α1R-NKA model were mainly positioned similar to the ligands obtained as a result of docking with α1S-NKA, but deepening of CTSs steroid cores in α1S- and α1R-NKA were remarkably different (Table 2, Figure 3E,F, Figure 4 and Appendix A). In the case of α1S-NKA, all three CTSs were more significantly buried in the transmembrane part of the protein in comparison with α1R-NKA (Table 2, Figure 3, Figure 4 and Appendix A). The predicted binding energy was lower in the complexes with the α1R-NKA model (Table 3). The number of contacts of CTSs with protein was higher in complexes with α1S-NKA than in the α1R-NKA model. Similar amino acid residues involved in interactions with CTSs were found in both α1S-NKA and α1R-NKA models (Table 4). It should be noted that after substitution of Gln111 and Asn122 in α1R-NKA with charged amino acids (Arg and Asp), amino acid residue 122 in CTS-resistant form did not participate in the interaction with the ligands in contrast to amino acid residue 111, which continue to interact with CTS.

Comparison of contacts of different CTSs with amino acids of α1S-NKA CTSs-binding site demonstrates the decrease in the total number of contacts in the range of ouabain, digoxin and marinobufagenin (18 for ouabain, 13 for digoxin and 7 for marinobufagenin) with the simultaneous change of relative contribution of hydrophobic contacts (9 for ouabain, 9 for digoxin and 3 for marinobufagenin; Table 4). Comparison of CTSs binding with α1R- and α1S-NKA binding sites revealed that all studied CTSs have contacts only with five amino acid CTSs-binding site residues of α1R-NKA.

## 3. Discussion

NKA from various tissues demonstrates different sensitivity to CTSs depending on the chemical structure of CTS, composition of the NKA subunits and the origin of the enzyme (rodent or other mammals) for the α1-isoform [31]. It was established earlier only for α1S-NKA that ouabain and digoxin are pseudo-irreversible inhibitors due to the slow dissociation of the CTS–NKA complex [70,71]. We confirmed this in our experiment, where inhibitory effects of ouabain and digoxin, which were preincubated with α1S-NKA in E2P-conformation, were preserved after the dilution of complex CTS–NKA (Figure 2C,D). However, the inhibitory effect of marinobufagenin disappeared after the dilution of its complex with NKA, demonstrating that this CTS binds to the α1S-NKA reversibly. It was surprising because CTSs are mainly irreversible inhibitors of NKA, and there was no information concerning the reversibility of the marinobufagenin effect in the literature up to now.

According to ITC data, the affinity of α1S-NKA in E2P-conformation for marinobufagenin is significantly lower than that for ouabain and digoxin (at 37 °C about 44 and 11 times, respectively). These results correspond to our data [64] obtained earlier by the same method at 25 °C for NKA from duck salt glands that also have α1S-isoform [65]. The binding of ouabain to NKA is an enthalpy-driven process that is characteristic for the binding due to the formation of hydrogen bonds and Van der Waals interactions. However, digoxin and marinobufagenin binding occurs with the lowering of enthalpy contribution, and simultaneously entropy factor contribution appears (Figure 1, Table 1). The decrease in enthalpy contribution, in this case, may be due to the lower number of hydroxyl groups of steroid core of these CTSs because many hydroxyl groups participate in the interaction of ouabain with NKA [72]. This suggestion also correlates with the modeling data that demonstrate the increase in the relative contribution of hydrophobic contacts of digoxin with the CTSs-binding site in comparison with ouabain (Table 4).

IC_50_ values for inhibition of NKA by ouabain and digoxin are higher than K_d_ evaluated for their binding to the NKA in E2P-conformation. At the same time, K_d_ and IC_50_ values for marinobufagenin are almost the same. It may be explained by the finding that the affinity of marinobufagenin for NKA in E2P and E1/E2 states (that exists during the activity measuring) nearly does not change, whereas ouabain affinity significantly decreases as a result of the transition from E2P to E1/E2 [64].

Kanai and co-authors were able to generate crystals of NKA in E2P-form with different eight bound CTSs [73]. According to the results of the 3D analysis that was carried out by these authors, CTSs bind deeply in the preformed cavity of the E2P-conformation, which is open to the extracellular side of the membrane. They demonstrated that the position of the CTSs’ steroid core is virtually the same for all studied CTSs (including ouabain, digoxin and bufalin) independently of the presence or absence of sugar moiety, variation in the lactone ring and modifications of the steroid core [73]. According to the data obtained as a result of CTSs docking to two structures of NKA in E2P-conformation (4HYT и 4RET), ouabain and digoxin are located in the binding site similarly, whereas marinobufagenin position in the site is more superficial (distance between C23 of ouabain or digoxin and marinobufagenin is about 5 Å; Appendix A).

There are no data concerning the 3D structure of NKA with bound marinobufagenin, but there is such a structure for its analog bufalin [67,73]. However, the localization of bufalin in the binding site of NKA significantly differs from that of marinobufagenin. According to the data of 3D analysis and our modeling (Figure 3D–F), the location of bufalin in the CTSs-binding site of NKA is the same as the location of ouabain or digoxin. A comparison of structures of these two CTSs (marinobufagenin and bufalin, see Appendix A) demonstrates that the difference between them is in the occurrence of the OH-group in the C14 position of bufalin (instead of oxide bridge between C14 and C15 in marinobufagenin) and the OH-group instead of H at C5. The appearance of the oxygen bridge between C14 and C15 seems to make the structure of the steroid core harder and results in the formation of a bend in the region of the D-ring that interferes with its deepening into the cavity of the site. Moreover, the OH-group at C14 was noted as a critical feature for CTSs’ high affinity to NKA [74]. Admittedly, K_d_ for bufalin binding to E2P-conformation is close to K_d_ for ouabain and digoxin (125 nM in comparison with 88 and 126 nM for ouabain and digoxin, respectively) [73]; at the same time, its value is higher for marinobufagenin (2.3 µM; Table 1). Bufalin was shown to be a more potent inhibitor of NKA from pig kidneys than ouabain and digoxin (IC_50_ measured for the enzyme from pig kidneys are 110, 1500, 1950 and 900 nM for bufalin, marinobufagenin, digoxin and ouabain, respectively; [75]). It is in accordance with our data: IC_50_ evaluated for these CTSs in our experiments are 1200, 2000 and 800 nM for marinobufagenin, digoxin and ouabain, respectively.

It should be noted that the dependence of the NKA inhibition on ouabain and digoxin concentrations obtained after their incubation with NKA in E2P-conformation is described by sigmoid with a Hill coefficient close to 2 (Figure 2C,D). In accordance with the data of Kanai and co-authors, the Hill coefficient for the binding of CTSs with sugar moieties to NKA in E2P-conformation is close to 2. Investigators proposed that it is an indication of the interaction between two α-subunits in the NKA oligomer α2β2 during the binding of CTSs with corresponding sites or slow conformation transition that is induced by the binding of CTSs [73]. We should note that interactions between CTSs-binding sites were proposed earlier on the basis of curvature of the Scatchard plot for ouabain binding [76,77].

Our data demonstrate that in contrast to ouabain and digoxin (and bufalin too), marinobufagenin binds to α1S-NKA in E2P-conformation reversibly, which correlates with its K_d_ value and more superficial location in the CTSs-binding site described by the model. The peculiarities of marinobufagenin binding to the CTSs-binding site of α1S-NKA explain its distinguished physiological effects. We noted above that marinobufagenin induces cell death at higher concentrations than ouabain despite their close IC_50_ values (that means these two CTSs had an equal effect on the transport function of NKA; [34]). Indeed, the marinobufagenin concentration changes correlate with numerous diseases, such as renal ischemia, chronic kidney disease, myocardial infarction and preeclampsia [7,9,10,11,12]. It is assumed that an increase in the level of marinobufagenin in these pathologies plays a protective role. Moreover, earlier, Fedorova and co-authors showed that marinobufagenin treatment of mice with Alzheimer’s disease (AD) significantly decreased the inflammatory marker interleukin-6 (IL-6) mRNA in the cortex, which was higher in the AD mice than in wild-type mice [23]. In addition, other CTS—ouabain—demonstrated protection against okadaic acid-induced neuronal cell damage. In AD models (in vitro and in vivo), this cardiotonic steroid-activated autophagy-lysosomal signaling and reduced the level of phosphorylated tau [78]. Thus, cardiotonic steroids, such as marinobufagenin, bind to Na,K-ATPase superficially and, therefore, do not cause toxic effects. It can be considered for combination therapy in the treatment of AD. Additionally, it is worth noting that the extracellular part of Na,K-ATPase is a target of beta-amyloid peptide; hence, it seems promising to study the effect of CTSs on the functioning of NKA in AD. A change of CTSs’ levels in the blood is associated with pathologies, such as preeclampsia and myocardial infarction, and affects the NKA activity in erythrocytes [13], which can influence erythrocyte function and the supply of oxygen to tissues. Different inhibition modes of NKA and binding parameters to NKA obtained for the ouabain, digoxin and marinobufagenin allow us to propose that they have different roles in physiological regulation.

In contrast to α1S-NKA from pig kidneys, the binding of ouabain and digoxin to α1R-NKA from rat kidneys is reversible. Furthermore, marinobufagenin does not inhibit this isoform at all, which seems to be explained by their inability to bind with this isoform supported by ITC data. It should be noted that in our experimental conditions, the association constants below 2 × 10^4^ M^−1^ cannot be measured by the ITC method. ITC data also demonstrate that ouabain affinity to α1R-NKA is 60-times less than to α1S-NKA, and the energetic profile of the binding reaction was, in this case, similar to that for marinobufagenin: in particular, the contribution of enthalpy factor in free energy of binding decreased and the entropy factor appeared; thereby, it demonstrates the decrease in the contribution of hydrogen bonds and Van der Waals interactions. The course of these events is (according to docking data) the substitutions of Gln111 and Asn122 that results in the formation of a hydrogen bond (Figure 3A) that restricts the deepening of CTSs into the cavity of the CTSs-binding site and significantly decreases the number of contacts of CTSs with their binding site (Table 4). We observed a similar restriction of marinobufagenin deepening into the cavity of the binding site, possibly resulting from the curvature of its steroid core as a consequence of the appearance of an oxygen bridge between C14 and C15 on the D-ring. As a result of these events leading to the change of the deepening of CTSs in the cavity of the CTSs-binding site, these ligands are located 8–9 Å closer to the membrane surface. Additionally, trypsinolysis in the presence of CTSs (ouabain, digoxin and marinobufagenin) induces various sets of proteolytic fragments in α1S- but not in α1R-NKA. In E1-conformation, marinobufagenin, in contrast to ouabain or digoxin, binding to α1S-NKA leads to the formation of other fragments and, consequently, promotes diverse conformational change in the enzyme [68].

Therefore, comparison of our data with the results of other authors elucidates that namely deepening of the location of CTSs’ steroid core in the cavity of the CTSs-binding site may be responsible for the absence of the marinobufagenin cytotoxic effect [34] in the case of α1S-NKA and the absence of ouabain’s cytotoxic effect [45] in the case of α1R-NKA. This type of CTSs’ location in the cavity of the binding sites appears to decrease the number of contacts with amino acid residues of the cavity and produce another NKA conformation that does not trigger a signal for cell death.

## 4. Materials and Methods

### 4.1. Na,K-ATPase Purification and Activity Measurements

Na,K-ATPase was purified from pig and rat kidney’s outer medulla according to the methods described by Jorgensen and Akayama, respectively [79,80]. The final pellets were solubilized in the solution containing 25 mM imidazole (pH 7.4), 1 mM EDTA and 0.25 M sucrose at a concentration of 5–10 mg/mL. After that, 50 μL aliquots of pellet suspension were stored at −80 °C. Protein concentration was determined by the method of Lowry [81] with bovine serum albumin as a standard.

### 4.2. Electrophoresis and Western Blot

Polyacrylamide gel electrophoresis in the presence of SDS was carried out according to the method of Laemmli [82]. Protein bends on the gels were stained with Coomassie Brilliant blue. In both preparations, mainly α1- (100 kDa) and β-subunit (56 kDa) were found (Appendix A). For Western blot analysis, proteins were transferred to the nitrocellulose membrane, which was blocked in PBS with 5% skimmed milk and 0.05% Tween-20 and incubated with anti-α1 Na,K-ATPase antibody (C464.6; Millipore, Temecula, CA, USA) diluted 1:2000 overnight at 4 °C. Subsequently, the membranes were treated with the horseradish peroxidase-conjugated secondary antibodies for 1 h at room temperature. The immunoreactivity was detected using the enhanced chemiluminescence SuperSignal™ West Femto Maximum Sensitivity Substrate kit (34095, ThermoFisher Scientific, Waltham, MA, USA) in accordance with the manufacturer’s instructions.

### 4.3. Hydrolytic Na,K-ATPase Activity Measurements

NKA activity was estimated as ATP cleavage using the enzyme coupled assay method [83]. Incubation medium (2 mL) contained 130 mM NaCl, 20 mM KCl, 4 mM MgCl_2_, 3 mM ATP, 30 imidazole (pH 7.4), 1 mM phosphoenolpyruvate, 0.2 mM NADH, 0.18 mM; NADH and pyruvate kinase (600–1000 units/mL)/lactate dehydrogenase (900–1400 units/mL). Before the start of the experiment, different concentrations of pyruvate kinase/lactate dehydrogenase were added to 0.6 μg/mL NKA, and the activity was measured to be sure that the activities of these two enzymes do not limit NKA activity. Incubation was carried out at 37 °C, and the concentration of Na,K-ATPase in the sample was 0.4–0.6 μg/mL. Specific activity of NKA was in the range of 1800–2000 and 500–750 μmol of ATP hydrolyzed/mg of protein per hour for pig and rat enzymes, respectively. To determine IC_50_, 2 μg NKA was preincubated with CTSs in the incubation medium for 10 min, and after that, the reaction was started by the addition of ATP.

To identify a mode of enzyme inhibition by CTSs after their binding to different enzyme conformations, we preincubated NKA in the medium containing 30 mM imidazole (pH 7.4), 3 mM MgCl_2_, 3 mM P_i_/Tris and 1 mM EDTA (E2P-conformation). Solutions of CTSs were made using 100% DMSO with a final concentration of ouabain 100 mM and digoxin and marinobufagenin at 10 mM. Then an aliquot of the corresponding stock solution was added to the preincubation medium to obtain a final CTSs concentration.

The scheme of the experiment is presented in Appendix A. After enzyme preincubation with different concentrations of CTSs for several minutes, the reaction was started by transferring the aliquot of the solution containing 0.25–2.5 μg of NKA from the preincubation medium into the incubation one. The concentration of CTSs was decreased by this procedure by 50–100 times. The reaction proceeded for 3–5 min, and the rate of NKA activity was constant during this time. The time of preincubation was determined as a minimal one that was sufficient for the achievement of a reaction rate that did not change with further increases in time. That time interval was equal to 10 min for E2P-conformation of pig and rat kidney enzymes. The control sample was preincubated under the same conditions with a concentration of DMSO that was added to the experimental medium with the corresponding CTS.

The plots describing the dependence of the NKA hydrolytic activity on CTSs concentration in the incubation medium were made using the Origin 8.1 program (OriginLab Corp., Northampton, MA, USA). They were fitted by hyperbola. The dependence of the Na,K-ATPase hydrolytic activity on the CTSs’ concentration in the preincubation media was fitted to the Hill 1 function:$$y=y_{0}+\left(y_{max}-y_{0}\right)\frac{x^{n}}{k^{n}+x^{n}},$$
where *y*—Na,K-ATPase activity; *x*—concentration of CTS; *k*—inhibition constant; *n*—Hill coefficient. Fitting was performed using Origin 8.1 software.

### 4.4. Isothermal Titration Calorimetry

The thermodynamic parameters of CTSs’ binding to the pig and rat NKA were measured using the MicroCal iTC200 instrument (MicroCal, Northampton, MA, USA), as described earlier [84,85]. α1S-NKA (pig) was transferred into E2P-conformation in the following way: enzyme preparations (5 mg) were suspended in 3 mL of buffer 10 mM imidazole, 3 mM Tris/P_i_ (pH 7.4), 1 mM EDTA, 0.1 mM DTT and 3 mM MgCl_2_ (E2P-conformation). DTT was added to the buffer immediately before the experiment. Suspensions of enzymes were centrifuged at 138,000 g for 90 min, and the pellets obtained were suspended in the buffer. α1R-NKA from rat kidneys was transferred into the buffer by dialysis.

Experiments were carried out at 37 °C. CTSs were diluted by a buffer that was used for enzyme suspension. Aliquots (2.5 μL) of ligands were injected into a 0.2 mL cell containing the protein solution to achieve a complete binding isotherm. The protein concentration on the cell ranged from 10 to 20 μM, and the ligand concentration in the syringe ranged from 50 to 200 μM. When we used digoxin or marinobufagenin, stock solutions were prepared using 100% DMSO, and then stock solutions were diluted to needed concentrations by the corresponding buffer; after that, aliquots were added to the cell. The medium with a protein of interest contained DMSO in the same concentration. Thus, the concentration of DMSO during the titration process did not change. Titration with ouabain was carried out with its solution in water and DMSO, and the titration curve was the same.

The resulting titration curves were fitted using the MicroCal Origin software, assuming one set of binding sites. Affinity constants (K_a_) and enthalpy variations (ΔH) were determined, and the Gibbs energy (ΔG) and entropy variations (ΔS) were calculated from the equation:ΔG = −RTlnK_a_ = ΔH – TΔS. 

### 4.5. Docking of Cardiotonic Steroids to Ouabain-Sensitive and Insensitive α1-Na,K-ATPase

Structures of porcine Na,K-ATPase in different conformations (E2P: 4HYT+ouabain (OBN); 4RET + digoxin (DGX)) were obtained from Protein Data Bank (rcsb.org). Models of ouabain-insensitive NKA were constructed in the Moe 2014.09 program (Chemical Computing Group Inc., Montreal, QC, Canada). In the structures, 4HYT and 4RET residues Gln111 and Asn122 were substituted to Arg111 and Asp122, respectively. Then, all atoms in the proximity of 4.5 Å from the ligands persisting in the structures were selected; ligands were removed; and all selected atoms were minimized in forcefield MMFF94x. Similarly, except for the substitution of amino acid residues, control models of porcine NKA were created. RMSD between the lactone ring and steroid core of OBN from the 4HYT structure and OBN (lactone ring and steroid core) from the model after local minimization (α1S-NKA-OBN) and between the lactone ring and steroid core DGX from the 4RET structure and DGX (lactone ring and steroid core) from the model after local minimization (α1S-NKA-DGX) were evaluated. Models of ouabain, marinobufagenin and digoxin were constructed in Moe 2014.09. Ligands (OBN, MBG, DGX) and receptors (α1S- and α1R-NKA) were prepared for docking using AutoDockTools [86]. Ligands were refined during the Autodock Vina run. Local docking was performed using AutoDock Vina [87]. Docking results were analyzed using AutoDockTools [86]. The exhaustiveness was set by default (exhaustiveness = 8). Only the results with correct glycoside group positioning were included in the analysis.

### 4.6. Statistical Analysis

Mean values and standard deviations were calculated for all experiments. Student t-criterion with Bonferroni correction for multiple comparisons was used for the determination of statistically significant differences between the association constants. Probability values less than 0.05 were considered significant.

### 4.7. Chemicals

The remaining chemicals were supplied by Thermo Fisher Scientific (Waltham, MA, USA), Millipore (Temecula, CA, USA), Calbiochem (La Jolla, CA, USA), Sigma-Aldrich (St. Louis, MO, USA), and Anachemia Canada Inc. (Montreal, QC, Canada).

## 5. Conclusions

In this study, we demonstrated differences between the interaction of ouabain, digoxin and marinobufagenin with α1S- and α1R-NKA. In α1R-NKA, the substitution of amino acid residues into Arg111 and Asp122 leads to the hydrogen bond formation that alters the depth of the penetration in the CTSs-binding site. Deepening of the location of CTSs’ steroid core in the cavity of the CTSs-binding site in NKA determines the binding parameters and inhibition mode of CTSs that may be responsible for the absence of the marinobufagenin cytotoxic effect in the case of α1S-NKA and absence of the ouabain cytotoxic effect in the case of α1R-NKA. In both cases, CTSs’ location in the cavity of the binding sites appears to decrease the number of contacts with amino acid residues of the cavity.

## Figures and Tables

**Figure 1 ijms-22-13268-f001:**
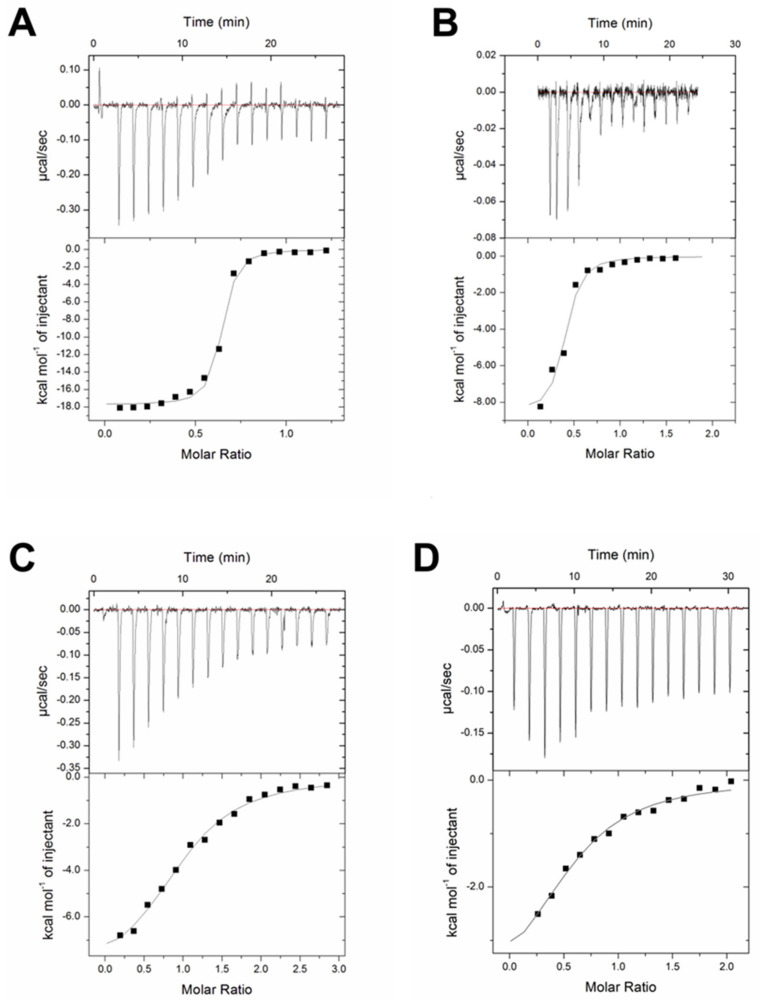
Cardiotonic steroids interaction with Na,K-ATPase from pig and rat kidneys. ITC data for ouabain (**A**), digoxin (**B**), and marinobufagenin (**C**) binding to α1S−Na,K-ATPase in E2P—conformation and ouabain (**D**) binding to α1R-Na,K-ATPase in E2P—conformation. Titration curves (upper panel) and binding isotherms (lower panel) are shown for the Na,K-ATPase interaction with ouabain, digoxin and marinobufagenin at 37 °C, pH 7.4. Approximation of the isotherm was made using a model with one type of binding site (continuous line).

**Figure 2 ijms-22-13268-f002:**
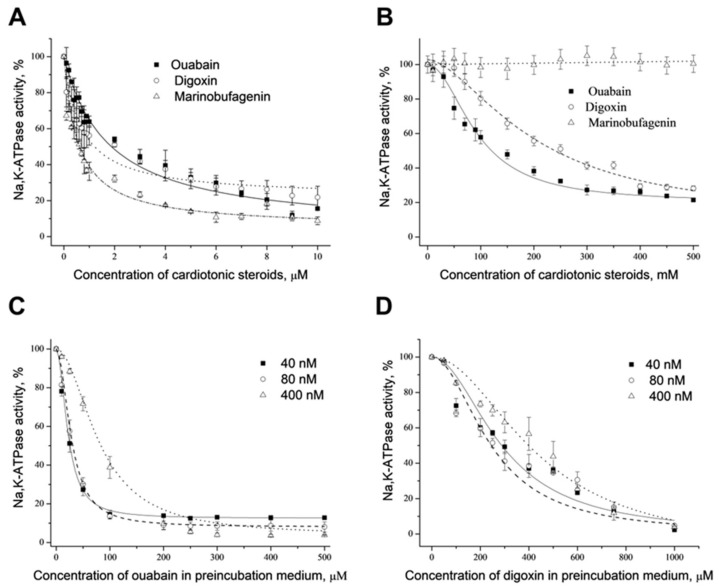
Dose-dependent effect of cardiotonic steroids on Na,K-ATPase activity from pig and rat kidneys. Dose-dependent inhibition of Na,K-ATPase activity from pig (**A**) and rat (**B**) kidneys by different cardiotonic steroids (ouabain, digoxin, marinobufagenin), which were added into incubation media. Dose-dependent “irreversible” inhibition of Na,K-ATPase from pig kidneys in E2P-conformation by ouabain (**C**) or digoxin (**D**). Cardiotonic steroids were added into preincubation media and stayed with the enzyme in E2P-conformation for the time that was necessary to reach a steady state. Then aliquots of preincubation medium (20 μL) were transferred into the incubation medium with a volume of 2 mL to assay the enzyme activity. For details, see “Materials and Methods”. Concentration of Na,K-ATPase: 1—20 nM; 2—80 nM; 3—400 nM. Means ± S.D. from 3 experiments performed in triplicates are shown.

**Figure 3 ijms-22-13268-f003:**
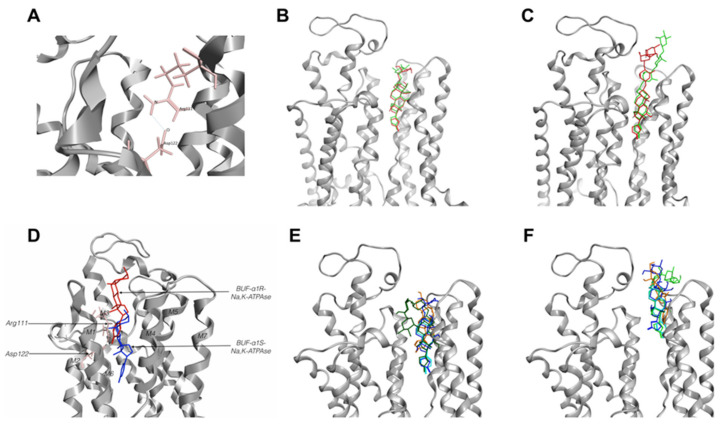
Cardiotonic steroids-binding site in Na,K-ATPase models. (**A**)—cardiotonic steroids (CTSs) binding site in α1R-Na,K-ATPase. Hydrogen bonds between Arg111 and Asp122 in α1R-Na,K-ATPase. (**B**,**C**)—verification of ouabain and digoxin position in CTSs-binding site of α1S-Na,K-ATPase. Comparison from X-ray 4HYT and 4RET structures with docking models: (**B**)—ouabain in 4HYT structure, (**C**)—digoxin in 4RET structure. CTSs from 4HYT or 4RET colored red, CTSs from docking models colored green. (**D**)—bufalin position in CTSs-binding site of α1S- (blue) and α1R-Na,K-ATPase (red). Models based on the 4HYT structure of Na,K-ATPase. (**E**,**F**)—CTSs positions in the CTSs-binding site of α1S-Na,K-ATPase (**E**) and α1R-Na,K-ATPase (**F**) for CTS. Bufalin—shown in turquoise, digoxin—green, ouabain—blue, marinobufagenin—orange. Models based on the 4HYT structure of Na,K-ATPase. BUF—bufalin.

**Figure 4 ijms-22-13268-f004:**
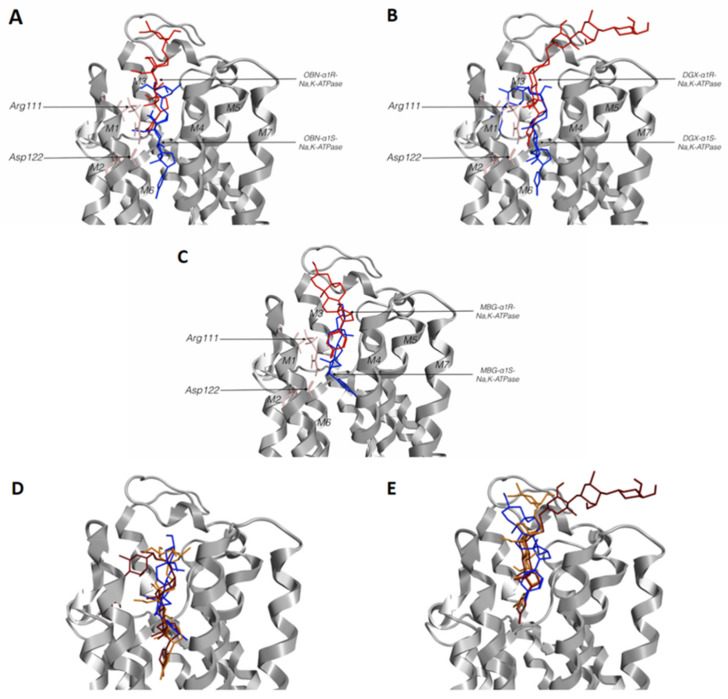
Superposition of cardiotonic steroids (CTSs) in α1S- and/or α1R-Na,K-ATPase docking models. CTSs’ positions in the CTSs-binding site of α1S and α1R-Na,K-ATPase for: (**A**)—ouabain, (**B**)—digoxin, (**C**)—marinobufagenin. Arg111 and Asp122 are colored in pink, CTSs interacting with α1R-Na,K-ATPase are colored red, with α1S-Na,K-ATPase colored in blue. (**D**)—Position of ouabain (orange), digoxin (red) and marinobufagenin (blue) in α1S-Na,K-ATPase. (**E**)—Position of ouabain (orange), digoxin (red) and marinobufagenin (blue) in α1R-Na,K-ATPase. Models based on the 4HYT structure of Na,K-ATPase. OBN—ouabain; DGX—digoxin; MBG—marinobufagenin.

**Table 1 ijms-22-13268-t001:** Thermodynamic parameters of α1S- and α1R-Na,K-ATPase in E2P-conformation binding to ouabain, digoxin and marinobufagenin (MBG) at pH 7.4 and 37 °C.

Na,K-ATPase	CTS	K_a_, M^−1^	K_d_, nM	n	ΔH, kcal/mol	TΔS, kcal/mol	ΔG, kcal/mol
α1S-Na,K-ATPase	ouabain	1.9 × 10^7^	53	0.7 ± 0.2	−21.6	−11.3	−10.3
	digoxin	4.0 × 10^6^ *	208	0.5 ± 0.1	−7.9	2.0	−9.9
	MBG	4.3 × 10^5^ *	2320	1.7 ± 0.2	−5.2	2.8	−7.9
α1R-Na,K-ATPase	ouabain	3.1 × 10^5^ *	3226	0.6 ± 0.1	−4.3	3.4	−7.7
	digoxin	nd		nd	nd		
	MBG	nd		nd	nd		

All measurements were performed three times. MBG—marinobufagenin; K_a_—association constant, standard deviation did not exceed ±20%; K_d_—dissociation constant; calculated as K_d_ = 1/K_a_; n—reaction stoichiometry; ΔH—enthalpy variation; standard deviation did not exceed ±20%; TΔS—entropy variation; standard deviation did not exceed ±20%; * *p* < 0.01 in comparison with α1S-Na,K-ATPase:ouabain complex K_a_; ΔG—Gibbs energy; Calculated from the equation: ΔG = −RTlnK_a_; nd—not detected.

**Table 2 ijms-22-13268-t002:** Difference in the deepening of the steroid nucleus of cardiotonic steroids between α1S- Na,K-ATPase and α1R-Na,K-ATPase. The distance was measured from C23 of the ligand docked to α1S- Na,K-ATPase (4HYT) to the C23 of the ligand docked to α1R- Na,K-ATPase (4HYT).

	OBN	DGX	MBG
α1S-NKA (4HYT) -α1R-NKA (4HYT)	9.173 Å	7.624 Å	7.446 Å
α1S-NKA (4RET)- α1R-NKA (4RET)	8.719 Å	8.994 Å	2.015 Å

Å—Angstrom; α1S-NKA—α1-subunit of a porcine Na,K-ATPase; α1R-NKA—α1-subunit of a rat Na,K-ATPase; OBN—ouabain; DGX—digoxin; MBG—marinobufagenin.

**Table 3 ijms-22-13268-t003:** Predicted binding energy of cardiotonic steroids with α1S- and α1R-Na,K-ATPase.

	OBN	DGX	MBG
Model	Predicted Binding Energy (kcal/mol)		
α1S-NKA (4HYT)	−11.0	−12.0	−9.0
α1R-NKA (4HYT)	−7.2	−8.0	−7.8
α1S-NKA (4RET)	−10.6	−11.8	−9.5
α1R-NKA (4RET)	−7.4	−9.0	−8.2

α1S-NKA—α1-subunit of a porcine Na,K-ATPase; α1R-NKA—α1-subunit of a rat Na,K-ATPase; OBN—ouabain; DGX—digoxin; MBG—marinobufagenin.

**Table 4 ijms-22-13268-t004:** Amino acid residues of α1S- and α1R-NaK-ATPase involved in the interaction with different cardiotonic steroids.

Cardiotonic Steroid	OBN		DGX		MBG	
4HYT						
**Model**	**α1S**	α1R	α1S	α1R	α1S	α1R
Amino acid residues	**Gln 111**	**Arg111**	Asp 121	Arg111	**Gln 111**	**Arg 111**
	**Glu 116**	**Glu 116**	Asn 122	Glu307	Asp 121	Ile 315
	Glu 117	**Phe 783**	Leu 125	Thr 309	Asn 122	**Phe 316**
	Asp 121	**Phe 786**	Glu 312	**Phe 783**	**Phe 316**	**Phe 783**
	Asn 122	Val 881	Ile 315	**Arg 880**	**Phe 783**	**Arg 880**
	Leu 125		Phe 316		Phe 786	
	Glu 312		Gly 319		Thr 797	
	Ile 315		Ala 323		**Arg 880**	
	Phe 316		**Phe 783**			
	Gly 319		Phe 786			
	Ala 323		Leu 793			
	**Phe 783**		Thr 797			
	**Phe 786**		Ile 800			
	Leu 793		**Arg 880**			
	Thr 797					
	Ile 800					
	Arg 880					
	Asp 884					
4RET						
Amino acid residues	**Gln111**	**Arg111**	**Gln111**	**Arg111**	**Gln111**	**Arg111**
	Glu116	**Glu117**	Thr114	Glu117	**Glu117**	Glu116
	**Glu117**	**Phe783**	Glu115	Glu312	Asp121	**Glu117**
	Pro118	Arg880	Asp121	**Ile315**	Asn122	Ile315
	Asn122	Asp884	Asn122	Phe316	Phe316	**Phe783**
	Leu125	Asp885	Leu125	**Phe783**	**Phe783**	
	Phe316		**Ile315**	Phe786	Arg880	
	Gly319		Gly319	Leu793		
	Ala323		Ala323	Asp885		
	Thr797		**Phe783**	Arg886		
	**Phe783**		Thr797	Thp887		
			Ile800			

Interacting with ligand amino acid residues repeating in CTSs-binding sites both in α1S- and α1R-Na,K-ATPase marked as bold. α1S—α1-subunit of a porcine Na,K-ATPase; α1R—α1-subunit of a rat Na,K-ATPase; OBN—ouabain; DGX—digoxin; MBG—marinobufagenin.

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
