# Peer review of "Depth of the Steroid Core Location Determines the Mode of Na,K-ATPase Inhibition by Cardiotonic Steroids"

_ijms, 2021, doi:10.3390/ijms222413268_

Round 1

Reviewer 1 Report

In this manuscript, Tverskoi and coworkers have used an in vitro/in silico approach to compare binding parameters and the inhibition mode of cardiotonic steroids (CTS), namely ouabain, digoxin, and marinobufagenin, to the Na/K-ATPase (NKA) sensitive (pig kidney – α1S-NKA) and resistant (rat kidney – α1R-NKA) α1 isoform. They have described differences in thermodynamic parameters of interaction of CTS and NKA, which agree with dose-response data and molecular docking findings. This work further improves our understanding of how different CTS can exert different effects through the ion-pumping or signaling function of NKA.

Comments:

  1. In the introduction section, lines 101-103, the authors should better explain that “discrepancy in cytotoxic action of CTS on human and rodent cells” is merely an extrapolation of the presented data and comparison with data from the scientific literature. There is no data on cell cytotoxicity presented in this work.
  2. Since it has been previously reported in the literature that digoxin and marinobufagenin are able to bind and inhibit the rodent NKA, do the authors have any explanation why they were not able to detect the thermodynamic parameters of those compounds using the α1R-NKA in the isothermal titration calorimetry experiments? Is that a limitation of this technique? The answers would be nicely fitted under the discussion section.
  3. The authors should improve their methodological description of the molecular docking for better clarity and reproducibility:
    • What method did they use for refinement of ligands?
    • What was the root mean square deviation (RMSD) obtained from the re-docking step after their methodological validation?
    • Did the authors set any exhaustiveness during the process to search for best conformers into the binding site?
  4. In the conclusion section, lines 496-498, the authors should avoid making extrapolations of different NKA conformations and cell death signaling. This statement does not come as a direct conclusion from this work, since there is no cell signaling experiments reported. Nevertheless, it would fit under the discussion section along with literature citations.
  5. Where did the Student t-criterion method was used as statistical analysis? The authors should report in the respective figure legend.
  6. Please define the cardenolides and bufadienolides in the Figure S1 legend.

Author Response

The authors express deep gratitude to the editors and reviewers for the interest in the present study, for a careful reading of the manuscript, constructive criticism, and valuable comments. The content of these comments and our responses to them are given below.

Response to the Reviewer 1 Comments:

Point #1: In the introduction section, lines 102-105, the authors should better explain that “discrepancy in cytotoxic action of CTS on human and rodent cells” is merely an extrapolation of the presented data and comparison with data from the scientific literature. There is no data on cell cytotoxicity presented in this work.

Response #1: We changed in the introduction section lines 102-105 (gray color):

Our results show that the distinction in deepening of CTS determines the differences between the binding parameters and the mode of NKA inhibition by diverse CTS. The data are useful for the understanding of our previous findings about the discrepancy between NKA conformational changes induced by various CTS [68] as well as between cytotoxic CTS action on human and rodent cells [45].

Point #2: Since it has been previously reported in the literature that digoxin and marinobufagenin are able to bind and inhibit the rodent NKA, do the authors have any explanation why they were not able to detect the thermodynamic parameters of those compounds using the α1R-NKA in the isothermal titration calorimetry experiments? Is that a limitation of this technique? The answers would be nicely fitted under the discussion section.

Response #2: Yes, ITC method limitations do not allow to detect digoxin and marinobufagenin interaction with the rodent NKA (containing α1-subunit). The shape of the isotherm is dependent on affinity constant (Ka) and the concentration of the interacting component in the calorimeter cell. The product of these two terms gives a number called the C value.

,

where n is stoichiometry of the interaction.

C value should be higher than 1 (for best results 10-100) [Ladbury, John E., and Michael L. Doyle, eds. Biocalorimetry 2: applications of calorimetry in the biological sciences. John Wiley & Sons, 2004.]. A lower Kavalue demands a high concentration of Na,K-ATPase in the cell. The Na,K-ATPase concentration after isolation and purification (according to Jørgensen method [79]) is about 10 mg/ml (~64 µM). Even if we use this protein concentration (without a dialysis dilution), we will not be able to measure the Ka value below 2 × 104 M-1.

We added in the discussion section lines 363-364 (gray color):

Furthermore, marinobufagenin does not inhibit this isoform at all, which seems to be explained by their inability to bind with this isoform supported by ITC data. It should be noted that in our experimental conditions the associationconstants below 2 × 104 M-1 cannot be measured by the ITC method.

Point #3: The authors should improve their methodological description of the molecular docking for better clarity and reproducibility:

  • What method did they use for refinement of ligands?
  • What was the root mean square deviation (RMSD) obtained from the re-docking step after their methodological validation?
  • Did the authors set any exhaustiveness during the process to search for best conformers into the binding site?

Response #3:

  • Ligands were refined during the Autodock Vina run. Each run can produce several results, which are merged, refined, clustered and sorted automatically to produce the final result [[87] (Trott O., Olson A. J. AutoDock Vina: improving the speed and accuracy of docking with a new scoring function, efficient optimization, and multithreading //Journal of computational chemistry. – 2010. – Т. 31. – â„–. 2. – P. 455-461)].

We added this information in the Materials and methods section lines 486-487 (gray color):

Ligands were refined during the Autodock Vina run.

  • We have measured root mean square deviation (RMSD) between the lactone ring and steroid core of OBN from the 4HYT structure and OBN (lactone ring and steroid core) from the model after local minimization (α1S-NKA-OBN). The same measurements were performed between DGX from the 4RET structure and DGX from the model after local minimization (α1S-NKA-DGX). RMSD for ouabain and digoxin were 0.57 Å and 0.73 Å, respectively.

We added data in the Results section lines 193-196 (gray color):

In both cases, docking with the minimized model of α1S-NKA results in the positioning of ouabain into the channel formed by M1-M5 helixes like ouabain from structure 4HYT (root mean square deviation (RMSD) - 0.57 Å for the lactone ring and steroid core), and a molecule of digoxin was in the channel similar to the ligand from structure 4RET (RMSD - 0,73 Å for the lactone ring and steroid core).

We added information about RMSD in the materials and methods section lines 480-484 (gray color):

RMSD between the lactone ring and steroid core of OBN from the 4HYT structure and OBN (lactone ring and steroid core) from the model after local minimization (α1S-NKA-OBN), and between the lactone ring and steroid core DGX from the 4RET structure and DGX (lactone ring and steroid core) from the model after local minimization (α1S-NKA-DGX) were evaluated.

  • The exhaustiveness was set by default (exhaustiveness=8).

We included the information in the Materials and methods section lines 488-489 (gray color):

The exhaustiveness was set by default (exhaustiveness=8).

Point #4: In the conclusion section, lines 496-498, the authors should avoid making extrapolations of different NKA conformations and cell death signaling. This statement does not come as a direct conclusion from this work, since there is no cell signaling experiments reported. Nevertheless, it would fit under the discussion section along with literature citations.

Response #4: We removed this statement from the conclusions section.

Point #5: Where did the Student t-criterion method was used as statistical analysis? The authors should report in the respective figure legend.

Response #5: We added information in the materials and methods section lines 493-495 (gray color):

Student t-criterion with Bonferroni correction for multiple comparisons was used for determination of statistically significant difference between the association constants. Probability values less than 0.05 were considered significant.

We added information in table 1 (gray color).

Point #6: Please define the cardenolides and bufadienolides in the Figure S1 legend.

Response #1: In the Figure S1 legend cardenolides and bufadienolides are shown (gray color).

Figure S1. Structures of cardiotonic steroids. Cardenolides - ouabain and digoxin, bufadienolides - marinobufagenin and bufalin are shown.

Reviewer 2 Report

In the article entitled "Depth of the steroid core location determines the mode of Na,K-ATPase inhibition by cardiotonic steroids" Tverskoi et al compared the inhibition mode of CTS (ouabian, digoxin, and marinobufagenin in cell free systems using CTS-sensitive (from pig kidney) and CTS-insenstive (from rat kidney) Na,K-ATPase alpha1 subunit. The results showed that marinobufagenin interacted reversibly while ouabain and digoxin interacted irreversibly with Na,K-ATPase alpha1 subunit. Although, the results are very interesting and could describe the differences in signaling through different cardiotonic steroids. The manuscript is very well written and the data are very important towards understanding the role of cardiotonic steroids in signaling. The only major comment I have is does this holds true in a physiological system.

The authors should show the reversibility of action of marinobufagenin in a cell culture model like in LLC-PK1 and NRK cells.

Author Response

Point: The authors should show the reversibility of action of marinobufagenin in a cell culture model like in LLC-PK1 and NRK cells.

Response: In this study, we demonstrated the binding reversibility of various CTS in vitro using purified Na,K-ATPase. We comprehensively agree with the reviewer’s advice to check marinobufagenin inhibitor type in cell experiments. To investigate the reversible action of marinobufagenin in cell culture models in the future we plan to order LLC-PK1 and NRK cell lines.

Round 2

Reviewer 2 Report

Well written manuscript. No further comments